# Self-Adaptable Point Processes with Nonparametric Time Decays

**Zhimeng Pan, Zheng Wang, Jeff M. Phillips, and Shandian Zhe**
School of Computing, University of Utah
Salt Lake City, UT 84112
{z.pan,wzhut}@utah.edu, jeffp@cs.utah.edu, zhe@cs.utah.edu

## Abstract

Many applications involve multi-type event data. Understanding the complex influences of the events on each other is critical to discover useful knowledge and to predict future events and their types. Existing methods either ignore or partially account for these influences. Recent works use recurrent neural networks to model the event rate. While being highly expressive, they couple all the temporal dependencies in a black-box and can hardly extract meaningful knowledge. More important, most methods assume an exponential time decay of the influence strength, which is over-simplified and can miss many important strength varying patterns. To overcome these limitations, we propose SPRITE, a Self-adaptable Point pRocess wIth nonparametric Time dEcays, which can decouple the influences between every pair of the events and capture various time decays of the influence strengths. Specifically, we use an embedding to represent each event type and model the event influence as an unknown function of the embeddings and time span. We derive a general construction that can cover all possible time-decaying functions. By placing Gaussian process (GP) priors over the latent functions and using Gauss-Legendre quadrature to obtain the integral in the construction, we can flexibly estimate all kinds of time-decaying influences, without restricting to any specific form or imposing derivative constraints that bring learning difficulties. We then use weight space augmentation of GPs to develop an efficient stochastic variational learning algorithm. We show the advantages of our approach in both the ablation study and real-world applications.

## 1 Introduction

Events of multiple types are ubiquitous, such as in online shopping, social networking and biological signal transduction. Understanding the complex influences of those events on each other, including excitation, inhibition, and how the strength of these influences varies with time, is crucial to discover useful knowledge and to predict future events and their types, which can benefit many applications, *e.g.*, marketing analysis, biological study, and early warnings of catastrophes.

While many excellent works have been proposed for event modeling and analysis (Blundell et al., 2012; Linderman and Adams, 2014; Xu et al., 2016; Tan et al., 2016; Xu and Zha, 2017; Zhang et al., 2020a), most of them are inadequate to capture complex influences between the events. The classical Poisson processes simply ignore these influences and assume independent increments. Although Hawkes processes (HP) can estimate the mutual excitation between the events, they ignore the inhibition effect, which is common in real world. More important, existing methods (Xu et al., 2016; Zhang et al., 2020a), even for those considering both the excitation and inhibition, mostly assume an exponential time decay of the influence strength, which is over-simplified and can miss many other decay patterns. While recent works have also used recurrent neural networks (RNNs) (Du et al., 2016; Mei and Eisner, 2017) to model the event rate and hence are highly expressive, they couple

all the temporal dependencies in a black box, and are difficult to distill meaningful and important knowledge, *e.g.*, influence types and strengths.

To address these limitations, we propose SPRITE, a multi-variate self-adaptable point process with nonparametric time decays. Our method not only can discover the type of the influence (*i.e.*, excitation or inhibition) between every pair of the events, but also is flexible enough to estimate all kinds of influence strength decaying patterns, not limited to the exponential decay or other particular forms. Specifically, we first introduce an embedding to represent each event type, and model the influence between any two events as an unknown function of the event type embeddings and time span. In so doing, we avoid separately estimating the influence between every pair of the events, and largely reduce the parameters. Next, we propose a general construction that covers all possible time decaying functions. We assign Gaussian process (GP) priors over the free latent functions and use Gauss-Legendra quadrature to calculate the integral in the construction. In this way, we can flexibly estimate a variety of time decaying influences while not needing to impose any derivative constraint, which can bring learning difficulties. For efficient inference, we use weight-space augmentation of GPs to prevent computing huge covariances matrices, and neural network feature mapping to enhance the learning capacity. We then use the re-reparameterization trick to develop a scalable stochastic variational learning algorithm.

For evaluation, we first examined SPRITE in an ablation study. We tested the case of a single event type with excitation effects only and three time decays, and a bi-type event case with mixed excitation and inhibition effects. In both cases, our method accurately recovered the rate function of each type of events and the influence between the events, performing better than state-of-the-art methods based on or extending Hawkes processes and RNNs. Next, we examined SPRITE in three real-world and one synthetic benchmark datasets. We examined the accuracy in predicting the occurrence time and type of future events. In both tasks, our method nearly always outperforms the competing approaches, often by a large margin. Finally, by looking into the embeddings and influence functions learned by our model, we found interesting patterns within the event types and their interactions.

## 2  Background

**Temporal Point Processes.**  We denote a sequence of events and their types by $\Gamma = [(t_1, s_1), \ldots, (t_M, s_M)]$, where each $t_i$ is the time point when the $i$-th event occurred ($t_1 \leq \ldots \leq t_M$), $s_i$ is the type of the $i$-th event. Suppose we have $K$ types of events in total; so each $s_i \in \{1, \ldots, K\}$.

Temporal point processes (TPPs) (Daley and Vere-Jones, 2007) provide a fundamental framework to characterize and model the events. A TPP is defined via the rate (or conditional intensity) function of the events, $\lambda_s(t)$. For example, a homogeneous Poisson process assumes each $\lambda_s(t)$ is a constant $\lambda_s^0$, and the past events $\{(t_j, s_j)|t_j < t\}$ do not have any influence on generating a new event at $t$. While being simple and convenient, Poisson processes completely ignore the interactions between the events. Hawkes processes (HPs) (Hawkes, 1971) overcome this issue by modeling the triggering effects between the events, $\lambda_s(t) = \lambda_s^0 + \sum_{t_j < t} \gamma_{s_j \to s}(t - t_j)$ where $\gamma_{s_j \to s}(\Delta) > 0$ is called "triggering kernel", and characterizes how largely the previous event at $t_j$ encourages a new event of type $s$ to occur at $t$. Naturally, this effect is supposed to decay with time $\Delta$. The most commonly used choice is the exponential decay,

$$\gamma_{s' \to s}(\Delta) = \alpha_{s's} \cdot \exp\left(-\tau_{s's}\Delta\right), \tag{1}$$

where $\alpha_{s's} > 0$ and $\tau_{s's} > 0$. Given the definition of $\lambda_s(\cdot)$, the probability of the event sequence is

$$p(\Gamma) = \prod_{k=1}^{K} \exp\left(-\int_0^T \lambda_k(t)\mathrm{d}t\right) \prod_{i=1}^{M} \lambda_{s_i}(t_i), \tag{2}$$

where $T$ is the total span of the events in $\Gamma$ and so $t_M \leq T$.

**RNN based Point Processes.** HPs only consider the triggering effects and hence can be quite restricted. In order to estimate arbitrarily varying rate functions, recent works Du et al. (2016); Mei and Eisner (2017) use recurrent neural networks (RNNs) to model the event dependencies and rates. The key idea is as follows. For each event $i$, we introduce a hidden state vector $\boldsymbol{\beta}(t_i)$. The event dependency is modeled through the updating rule of the hidden states,

$$\boldsymbol{\beta}(t_{i+1}) = \rho\left(\boldsymbol{\beta}(t_i), t_{i+1}\right) \tag{3}$$

where $\rho(\cdot)$ is a complicated mapping, fulfilled by a combination of neural network operations, such as linear transformation, nonlinear activation, and gating. Note that the event type $s_i$ can also be incorporated into $\rho(\cdot)$. The rate function is then defined as a transformation of $\boldsymbol{\beta}(t)$. We can therefore evaluate and maximize the point process likelihood (2) for model estimation.

## 3 Model

While RNN based methods are highly expressive and can capture arbitrarily complex dependencies, due to the black-box updating rule (see (3)), they can hardly distill meaningful knowledge, such as the influence type and strength between the events, which are critical for data analysis and knowledge discovery. On the other hand, while HPs enjoy excellent interpretability due to the additive structure in the intensity, they overlook the inhibition effect among the events, which is common in real world. More important, nearly all the HP based methods adopt exponential time-decaying kernels (or a combination of them) (see (1)), including the recent works (Mei and Eisner, 2017; Zhang et al., 2020a) that extend HPs to allow a negative triggering kernel ($\alpha_{s's} < 0$ in (1)). However, an exponential decay can be overly simplistic for various, complex real-world applications. These methods can therefore miss other important decaying patterns (*e.g.*, polynomial decays), resulting in inferior influence estimation and predictions.

To overcome these issues, we propose SPRITE, a multi-variate self-adaptable point process model that can explicitly estimate the influence type and strength between every pair of the events, and flexibly capture all kinds of time decays of the influence strength, not limited to a particular form. Specifically, we first define a raw rate function for each event type $s(1 \le s \le K)$ to characterize both the excitation and inhibition effects from the previous events,

$$\tilde{\lambda}_s(t) = \lambda_s^0 + \sum_{i=1}^{M} \mathbb{1}(t_i < t) h_{s_i \to s}(t - t_i), \tag{4}$$

where $\mathbb{1}(\cdot)$ is an indicator function, $\lambda_s^0$ is the background (or base) rate, and $h_{s_i \to s}(\cdot)$ is an influence term that characterizes the effect of a previous event of type $s_i$ happened at $t_i$ on the occurrence of a new event of type $s$ at $t$. When $h_{s_i \to s} > 0$, it indicates a triggering effect while $h_{s_i \to s} < 0$ means inhibition. The absolute value of $h_{s_i \to s}$ is the strength of the influence. Since the raw rate can be negative, we then apply a scaled soft-plus transformation (Mei and Eisner, 2017) to ensure we obtain a positive rate function, $\lambda_s(t) = \beta \log \left( 1 + \exp(\tilde{\lambda}_s(t)/\beta) \right)$ where $\beta > 0$.

Next, we introduce a set of embeddings $\mathcal{U} = \{\mathbf{u}_s\}_{s=1}^{K}$ to represent the event types. We sample these embeddings from the standard Gaussian distribution, $p(\mathcal{U}) = \prod_s \mathcal{N}(\mathbf{u}_s | \mathbf{0}, \mathbf{I})$. Then we model the influence term as a function of the event type embeddings and time span,

$$h_{s_i \to s}(\Delta) = h(\mathbf{u}_{s_i}, \mathbf{u}_s, \Delta). \tag{5}$$

The advantage of doing so is that we only need to estimate one function $h$ to capture the influence among all the events (including those within the same type). The embeddings can further enable us to discover the hidden structures of the event types (*e.g.*, cluster and outliers). Otherwise, if we estimate one function for each pair of event types, we have to learn $K(K + 1)/2$ functions, which can be much more challenging and costly, especially when the event types are many (*i.e.*, large $K$).

We want our model to flexibly learn the influence function $h(\cdot)$ and its strength decaying patterns with time, rather than restrict to any pre-specified, parametric form, *e.g.*, the exponential decay in (1). In other words, we only require when $h > 0$, the time derivative $\frac{\partial h}{\partial \Delta} < 0$, and when $h < 0$, $\frac{\partial h}{\partial \Delta} > 0$, *i.e.*, $h \cdot \frac{\partial h}{\partial \Delta} < 0$, to ensure $|h|$ decreases over time. While it is standard to assign a Gaussian process (GP) prior (Rasmussen and Williams, 2006) over $h$ to enable a flexible, nonparametric function estimation, incorporating the time derivative constraints is difficult. The derivative of a GP is still a GP (provided the kernel is differentiable). Since the support of the Gaussian distribution (GP finite projection) is the entire real space, imposing a monotonic constraint is infeasible (there are always nonzero probabilities for positive/negative derivatives). Existing work (Riihimäki and Vehtari, 2010) introduces a set of virtual points and encourage the derivative at these points to be as positive (or negative) as possible. This essentially is a soft regularization, not an actual constraint.

To address this issue, we observe that the time-decaying influence functions have a general structure.

**Lemma 3.1.** $h \cdot \frac{\partial h}{\partial \Delta} < 0$ *if and only if $h$ has the form*

$$h(\mathbf{u}_{s_i}, \mathbf{u}_s, \Delta) = g(\mathbf{u}_{s_i}, \mathbf{u}_s)(1 - f(\mathbf{u}_{s_i}, \mathbf{u}_s, \Delta)), \tag{6}$$

*where $g(\mathbf{u}_{s_i}, \mathbf{u}_s) = h(\mathbf{u}_{s_i}, \mathbf{u}_s, 0)$, $0 \le f(\cdot) < 1$, $\frac{\partial f}{\partial \Delta} > 0$, and $f(\mathbf{u}_{s_i}, \mathbf{u}_s, 0) = 0$.*

We leave the proof in the Appendix. Next, to construct $f$, we represent $f(\cdot) = \tanh\left(z(\mathbf{u}_{s_i}, \mathbf{u}_s, \Delta)\right)$ where $z(\cdot)$ is another function. To satisfy the requirements for $f$ in (6), we need to ensure $z \ge 0$, $z(\mathbf{u}_{s_i}, \mathbf{u}_s, 0) = 0$, and $\frac{\partial z}{\partial \Delta} > 0$. According to these constraints, we can represent $z$ as an integral function, $z(\mathbf{u}_{s_i}, \mathbf{u}_s, \Delta) = \int_0^\Delta \exp\left(\eta(\mathbf{u}_{s_i}, \mathbf{u}_s, r)\right) \mathrm{d}r$ where $\eta(\cdot)$ is a free function. Combining with (6), we now derive a general construction of our influence function,

**Corollary 3.1.1.** $h \cdot \frac{\partial h}{\partial \Delta} < 0$ *if and only if*

$$h(\mathbf{u}_{s_i}, \mathbf{u}_s, \Delta) = g(\mathbf{u}_{s_i}, \mathbf{u}_s)\left(1 - \tanh\left(\int_0^\Delta \exp\left(\eta(\mathbf{u}_{s_i}, \mathbf{u}_s, r)\right) \mathrm{d}r\right)\right) \tag{7}$$

*where both $g(\cdot)$ and $\eta(\cdot)$[1] are free latent functions.*

Note that our construction (7) also covers the special case that $h(\cdot) = 0$ (when $g = 0$), *i.e.*, no influence is from the previous event. To flexibly estimate $h(\cdot)$, we can place GP priors over $g$ and $\eta$. However, we no longer need to incorporate any derivative constraint and hence the learning is much easier. Note that the prior of $h(\cdot)$ is not GP anymore.

One might be concerned that the integral in (7) is analytically intractable and will bring troubles in model estimation. However, since the integration is one dimensional, we can use Gauss-Legendre quadrature to calculate the integral quite accurately and evaluate $h(\cdot)$. We will discuss the details in Sec. 4.2.

## 4 Algorithm

The model estimation is challenging. The GP prior over the latent function $g(\cdot)$ in (7) requires us to compute a multivariate Gaussian distribution of $\{g(\mathbf{u}_s, \mathbf{u}_k)\}_{1 \le s, k \le K}$ (the finite projection of $g$), and over $\eta(\cdot)$ a multivariate Gaussian distribution of $\{\eta(\mathbf{u}_s, \mathbf{u}_k, \Delta)\}_{1 \le s, k \le K, \Delta \in \mathcal{A}}$ where $\mathcal{A}$ are the time differences between every pair of events. Hence, even with a moderate number of events and event types (*e.g.*, $K = 100$), the corresponding covariance (kernel) matrices will be huge and infeasible to calculate. To address this problem, we use the weight space view of GPs (Rasmussen and Williams, 2006). That is, a GP model is equivalent to Bayesian linear regression after a (nonlinear) feature mapping. We use neural networks to construct a finite yet highly expressive mapping, and explicitly estimate the posterior of the (augmented) weight vector. In so doing, we can ease the inference and avoid computing the full covariance matrices. We then use Gauss-Legendre quadrature and the reparameterization trick to develop an efficient mini-batch stochastic variational learning algorithm.

### 4.1 GP Weight-Space Augmentation with Neural Network Feature Mapping

Specifically, we introduce $R$ dimensional weight vectors $\mathbf{w}_g$ and $\mathbf{w}_\eta$ for the latent functions $g(\cdot)$ and $\eta(\cdot)$ in (7). We then sample $g$ and $\eta$ in the following way:

$$\mathbf{w}_g \sim \mathcal{N}(\mathbf{w}_g|\mathbf{0}, \mathbf{I}), \quad \mathbf{w}_\eta \sim \mathcal{N}(\mathbf{w}_\eta|\mathbf{0}, \mathbf{I}),$$
$$g(\mathbf{u}_s, \mathbf{u}_k) = \mathbf{w}_g^\top \boldsymbol{\phi}_g(\mathbf{u}_s, \mathbf{u}_k), \quad \eta(\mathbf{u}_s, \mathbf{u}_k, \Delta) = \mathbf{w}_\eta^\top \boldsymbol{\phi}_\eta(\mathbf{u}_s, \mathbf{u}_k, \Delta), \tag{8}$$

where $\boldsymbol{\phi}_g$ and $\boldsymbol{\phi}_\eta$ are nonlinear feature mappings fulfilled by neural networks (NNs). It is known that if we marginalize out the weight vectors, we recover the GP model, where the function values at an arbitrary finite set of inputs follow a multivariate Gaussian distribution. The covariance (or kernel) function is an inner-product of the mapped feature vector. For example, the kernel for $g(\cdot)$ will be $\kappa_g([\mathbf{u}_s, \mathbf{u}_k], [\mathbf{u}_{s'}, \mathbf{u}_{k'}]) = (\boldsymbol{\phi}_g(\mathbf{u}_s, \mathbf{u}_k))^\top \boldsymbol{\phi}_g(\mathbf{u}_{s'}, \mathbf{u}_{k'})$; the NN parameters in $\boldsymbol{\phi}_g$ can be viewed as kernel parameters. Although the kernel does not correspond to an infinite feature mapping as in traditional kernels (*e.g.*, RBF), it still can be highly expressive due to the capacity of neural networks. We will keep the weight vectors $\mathbf{w}_g$ and $\mathbf{w}_\eta$ in our model inference. In so doing, we never need to operate the huge full covariance matrices. The dimension of the weight vectors $R$ is often set to be small, *e.g.*, 16 or 32. Hence the computation will be much more efficient, and the inference are easier and convenient.

---

[1]Rigorously speaking, $\eta$ should be exponential integrable, *i.e.*, $\exp(\eta(\cdot))$ is integrable in $[0, \Delta]$. This is a pretty mild condition, which can be satisfied by, *e.g.*, continuity of $\eta(\cdot)$.

Suppose we have observed a collection of $N$ independent event sequences, $\mathcal{D} = \{\Gamma^n\}_{1 \leq n \leq N}$ where each $\Gamma^n = \left[(t_1^n, s_1^n), \ldots, (t_{M_n}^n, s_{M_n}^n)\right]$ and $M_n$ is the number of events in sequence $n$. According to (2) and (8), the joint probability of our model is given by

$$p(\mathcal{U}, \mathbf{w}_g, \mathbf{w}_\eta, \mathcal{D}) = \prod_s \mathcal{N}(\mathbf{u}_s | \mathbf{0}, \mathbf{I}) \mathcal{N}(\mathbf{w}_g | \mathbf{0}, \mathbf{I}) \mathcal{N}(\mathbf{w}_\eta | \mathbf{0}, \mathbf{I})$$

$$\cdot \prod_{n=1}^{N} \prod_{k=1}^{K} \exp\left( -\int_0^{T_n} \lambda_k(t) \mathrm{d}t \right) \prod_{i=1}^{M_n} \lambda_{s_i^n}(t_i^n), \tag{9}$$

where $T_n$ is the total time span across the events in sequence $n$.

## 4.2 Stochastic Variational Learning

Given the joint probability (9), we aim to estimate the posterior distribution of the weight vectors $\mathbf{w}_g$ and $\mathbf{w}_\eta$, the event type embeddings $\mathcal{U}$, the parameters in the feature mappings $\phi_g$ and $\phi_\eta$, and the other parameters. Due to the intractable model evidence (*i.e.*, normalizer), exact inference is infeasible. Therefore, we use variational inference (Wainwright and Jordan, 2008). Specifically, we introduce a Gaussian variational posterior for the weight vectors,

$$q(\mathbf{w}_g, \mathbf{w}_\eta) = q(\mathbf{w}_g)q(\mathbf{w}_\eta) = \mathcal{N}(\mathbf{w}_g | \boldsymbol{\mu}_g, \boldsymbol{\Sigma}_g) \mathcal{N}(\mathbf{w}_\eta | \boldsymbol{\mu}_\eta, \boldsymbol{\Sigma}_\eta). \tag{10}$$

We use the Cholesky decomposition to parameterize the posterior covariance matrices to ensure their positive definiteness, $\boldsymbol{\Sigma}_g = \mathbf{L}_g \mathbf{L}_g^\top$ and $\boldsymbol{\Sigma}_\eta = \mathbf{L}_\eta \mathbf{L}_\eta^\top$ where $\mathbf{L}_g$ and $\mathbf{L}_\eta$ are lower triangular matrices. We then derive a variational model evidence lower bound (ELBO),

$$\mathcal{L} = -\mathrm{KL}\left(q(\mathbf{w}_g) \| p(\mathbf{w}_g)\right) - \mathrm{KL}\left(q(\mathbf{w}_\eta) \| p(\mathbf{w}_\eta)\right) + \log(p(\mathcal{U}))$$

$$+ \sum_{n=1}^{N} \left( -\sum_{k=1}^{K} \mathbb{E}_q \left[ \int_0^{T_n} \lambda_k(t) \mathrm{d}t \right] + \sum_{i=1}^{M_n} \mathbb{E}_q \left[ \log \left( \lambda_{s_i^n}(t_i^n) \right) \right] \right),$$

where $p(\mathbf{w}_g)$ and $p(\mathbf{w}_\eta)$ are the standard Gaussian priors in (8), and $p(\mathcal{U}) = \prod_s \mathcal{N}(\mathbf{u}_s | \mathbf{0}, \mathbf{I})$. To handle large $N, K$ (*i.e.*, the number of sequences and event types) and the intractable integration $\int_0^{T_n} \lambda_k(t) \mathrm{d}t$, we resort to stochastic optimization. Specifically, we partition all the event sequences of into mini-batches of size $B$: $\{\mathcal{B}_1, \ldots, \mathcal{B}_{N/B}\}$, and event types into mini-batches of size $C$: $\{\mathcal{C}_1, \ldots, \mathcal{C}_{K/C}\}$. We observe that the integration of the rate function is piece-wise, $\int_0^{T_n} \lambda_k(t) \mathrm{d}t = \sum_{j=0}^{M_n} \int_{t_j^n}^{t_{j+1}^n} \lambda_k(t) \mathrm{d}t$ where $t_0^n = 0$ and $t_{M_n+1}^n = T_n$. The rate $\lambda_k(t)$ in each interval $[t_j^n, t_{j+1}^n]$ is smooth (see (4)), and hence we use Gauss-Legendre quadrature to compute each $\int_{t_j^n}^{t_{j+1}^n} \lambda_k(t) \mathrm{d}t$. We now talk about how to compute a sample of $\lambda_k(t)$ for stochastic optimization, which uses the Gauss-Legendre quadrature again. Specifically, we arrange the ELBO as $\mathcal{L} = -\mathrm{KL}\left(q(\mathbf{w}_g) \| p(\mathbf{w}_g)\right) - \mathrm{KL}\left(q(\mathbf{w}_\eta) \| p(\mathbf{w}_\eta)\right) + \log(p(\mathcal{U})) - \sum_m \frac{B}{N} \sum_{n \in \mathcal{B}_m} \frac{N}{B} \sum_l \frac{C}{K} \sum_{k \in \mathcal{C}_l} \frac{K}{C} \mathbb{E}_q \left[ \int_0^{T_n} \lambda_k(t) \mathrm{d}t \right] + \sum_m \frac{B}{N} \sum_{n \in \mathcal{B}_m} \frac{N}{B} \sum_{i=1}^{M_n} \mathbb{E}_q \left[ \log \left( \lambda_{s_i^n}(t_i^n) \right) \right]$, which can be further viewed as an expectation,

$$\mathcal{L} = \mathbb{E}_{p(m), p(l)}[\hat{\mathcal{L}}_{m,l}], \quad \hat{\mathcal{L}}_{m,l} = -\mathrm{KL}\left(q(\mathbf{w}_g) \| p(\mathbf{w}_g)\right) - \mathrm{KL}\left(q(\mathbf{w}_\eta) \| p(\mathbf{w}_\eta)\right) + \log(p(\mathcal{U}))$$

$$- \frac{N}{B} \frac{K}{C} \sum_{n \in \mathcal{B}_m} \sum_{k \in \mathcal{C}_l} \mathbb{E}_q \left[ \int_0^{T_n} \lambda_k(t) \mathrm{d}t \right] + \sum_{n \in \mathcal{B}_m} \frac{N}{B} \sum_{i=1}^{M_n} \mathbb{E}_q \left[ \log \left( \lambda_{s_i^n}(t_i^n) \right) \right],$$

where $p(m) = \frac{B}{N}$, $m \in \{1, \ldots, \frac{N}{B}\}$, $p(l) = \frac{C}{K}$, $l \in \{1, \ldots, \frac{K}{C}\}$. To conduct efficient stochastic optimization, each step we first draw a mini-batch $\mathcal{B}_m$ and $\mathcal{C}_l$ to calculate the $\hat{\mathcal{L}}_{ml}$. We then use the reparameterization trick to draw from $q(\mathbf{w}_g, \mathbf{w}_\eta)$: $\hat{\mathbf{w}}_g = \boldsymbol{\mu}_g + \mathbf{L}_g \boldsymbol{\epsilon}_g$ and $\hat{\mathbf{w}}_\eta = \boldsymbol{\mu}_\eta + \mathbf{L}_\eta \boldsymbol{\epsilon}_\eta$, where $\boldsymbol{\epsilon}_g \sim \mathcal{N}(\cdot | \mathbf{0}, \mathbf{I})$ and $\boldsymbol{\epsilon}_\eta \sim \mathcal{N}(\cdot | \mathbf{0}, \mathbf{I})$. We substitute these samples into the corresponding terms in the expectations and obtain an unbiased stochastic estimate of $\hat{\mathcal{L}}_{ml}$. We then calculate the gradient to obtain an unbiased estimate of $\nabla \hat{\mathcal{L}}_{ml}$, which is also an unbiased estimate of $\nabla \mathcal{L}$. Accordingly, we can perform stochastic optimization to maximize $\mathcal{L}$ so as to estimate $q$ and the other parameters.

However, a critical issue is that after we use the parameterized samples of $\mathbf{w}_g$ and $\mathbf{w}_\eta$ to obtain the latent functions $g(\cdot)$ and $\eta(\cdot)$, we cannot compute the influence function $h(\cdot)$ in the rate function

$\lambda_k(\cdot)$, due to the intractable integral in (7). To address this issue, we observe that the integration is only one dimensional over time. Hence, we can use Gauss Legendre quadrature to calculate the integral with an analytical form,

$$\int_0^\Delta \exp\left(\eta(\mathbf{u}_{s_i}, \mathbf{u}_s, r)\right) \mathrm{d}r \approx \sum_j \omega_j \frac{\Delta}{2} \exp\left(\eta(\mathbf{u}_{s_i}, \mathbf{u}_s, \frac{\Delta}{2}\xi_j + \frac{\Delta}{2})\right), \tag{11}$$

where $\{\omega_j\}$ and $\{\xi_j\}$ are quadrature weights and nodes respectively. Note that this is based on the integral transform of the standard Gauss Legendre quadrature, which requires that the integration interval is $[-1, 1]$. We provide more details in the Appendix.

### 4.3 Algorithm Complexity

The time complexity of our inference algorithm is $\mathcal{O}(BCM + R^3)$ where $B$ and $C$ are the mini-batch sizes of the partitions of the event sequences and types, $M$ is the maximum length of the sequences, and $R$ is the dimension of weight vectors $\mathbf{w}_g$ and $\mathbf{w}_\eta$. Therefore, the computational cost is proportional to the mini-batch sizes, rather than determined by the total number of sequences $N$ and events types $K$. The space complexity is $\mathcal{O}(Kd + 2R^3)$, which is to store the covariances of $q(\mathbf{w}_g)$ and $q(\mathbf{w}_\eta)$, and the event type embeddings $\mathcal{U}$ ($d$ is the dimension of the embeddings).

## 5 Related Work

Many works use Poisson processes to analyze event data for its elegance and convenience, *e.g.*, (Lloyd et al., 2015; Schein et al., 2015, 2016, 2019). However, Poisson processes assume independent increments in their counting processes and hence overlook the interactions between the events. Hawkes processes (HPs) (Hawkes, 1971) therefore become popular, due to their capability of capturing the mutual excitations between the events. Many works propose HP based models to estimate the temporal relationships from data, *e.g.*, (Blundell et al., 2012; Tan et al., 2016; Linderman and Adams, 2014; Du et al., 2015; He et al., 2015; Wang et al., 2017; Yang et al., 2017a; Xu and Zha, 2017; Xu et al., 2018). Several methods have also been proposed to improve the inference of HPs, such as nonparametric triggering kernel (and base rate) estimation (Zhou et al., 2013; Zhang et al., 2020b; Zhou et al., 2020), Granger causality (Xu et al., 2016), short doubly-censored event sequences (Xu et al., 2017) and online estimation (Yang et al., 2017b). Another recent line of research (Zhe and Du, 2018; Pan et al., 2020; Wang et al., 2020) uses or extends HPs to learn the representations of event participants.

To overcome the limitation of HPs in only estimating the triggering effects, recent research efforts have been made to capture more complex temporal dependencies, *e.g.*, inhibition. In (Mei and Eisner, 2017), a simple extension of HPs was proposed to allow a negative coefficient in the exponential triggering kernel (*i.e.*, $\alpha_{s's} < 0$ in (1)), and the intensity was transformed through a soft-plus function to ensure positiveness. Zhou et al. (2021) uses a similar extension, but it models the kernel by a mixture of shifted Beta densities, and then uses a sigmoid transform scaled by an upper-bound to obtain a positive rate. The second method in (Mei and Eisner, 2017) develops a continues-time Long Short-Term Memory (LSTM) (Hochreiter and Schmidhuber, 1997) model to capture the event dependencies through the nonlinear, black-box updates of the hidden states. The rate is a transform of the hidden state. Du et al. (2016) developed a discrete RNN to model each event sequence. The rate at any time point $t$ is obtained by taking the exponential of a linear combination of the hidden state of the closest event, and the time span from that event. In addition, Omi et al. (2019) used feed-forward NNs to model the cumulative intensity function given the RNN states, so as to avoid computing the integration in the point process likelihood (see (2)), which is usually intractable. Shchur et al. (2019) modeled the time span between the events with a mixture of log-normal distribution parameterized by RNN states and other parameters.

Despite the expressive power of RNN based models, they encode all the temporal dependencies into the hidden state vectors and blackbox updating mechanisms. We can hardly extract important knowledge including the influence type and strength between a pair of events, how they vary with time, *etc*. The very recent work (Zhang et al., 2020a) uses the self-attention mechanism (Vaswani et al., 2017; Bahdanau et al., 2014) to capture the correlation between the events. The similar idea appears in the concurrent work (Zuo et al., 2020). Zhang et al. (2020a) modeled the rate as the soft-plus transformation of a base rate plus an exponential time-decaying kernel, where the base

rate, the amplitude and decaying rate are computed via the attention mechanism, which essentially calculates a weighted similarity with the representation of the past events. Although the model is more interpretable and highly expressive, like most HP models, it adopts an exponential time-decay for the influence strength, and hence can miss other decaying patterns in data. This (partly) motivates us to develop an interpretable yet flexible point process model to capture various time decays from data. There are other temporal models based on ODEs, *e.g.*, (Rubanova et al., 2019).

## 6 Experiment

### 6.1 Synthetic Data

To confirm the effectiveness of SPRITE, we first conducted an ablation study with synthetic data. Specifically, we considered two cases: single-type events and bi-type events. For single-type events, we assumed that they only have excitation effects on each other. We examined three time-decaying excitations,

$$
\begin{aligned}
h_1(\Delta) &= \max(0.4 - 0.1\Delta^2, 0), \quad (\textit{Quadratic}) \\
h_2(\Delta) &= \max(0.4 - 0.2\Delta, 0), \quad (\textit{Linear}) \\
h_3(\Delta) &= 0.3\exp(-0.8\Delta). \quad (\textit{Exponential})
\end{aligned}
\tag{12}
$$

We applied the scaled soft-plus transformation to ensure a positive rate. We set the scale $\beta = 0.8$. We used Thinning algorithm Lewis and Shedler (1979) to sample event sequences. For each case, we generated $10K$ sequences for training and $1K$ for validation. The length of each sequence is $32$.

For bi-type events, we assume type $0$ events trigger type $1$ events while type $1$ inhibits type $0$, and no effects are between events of the same type:

$$
h_{0\to1}(\Delta) = \max(1.0 - 0.05\Delta^2, 0), \quad h_{1\to0}(\Delta) = -0.5\exp(-0.5\Delta).
\tag{13}
$$

We set $\beta = 0.4$ and used scaled soft-plus to obtain the positive rate function. Again, we sampled 10K sequences for training and 1K for validation. Each sequence includes $64$ events.

We compared with the following state-of-the-art methods: (1) Hawkes processes (HP) with the standard exponential decaying triggering kernel, (2) Recurrent Marked Temporal Point Processes (RMTPP) (Du et al., 2016) that use a discrete RNN to model the event sequences, and use hidden RNN states and other parameters to estimate an overall rate function (for all types of events), (3) Neural Hawkes Processes (NeuralHP) (Mei and Eisner, 2017) that use a continuous LSTM to model the event sequences, and estimate the rate function for each event type, and (4) Self-Attentive Hawkes Process (SAHP) (Zhang et al., 2020a) that uses the self-attention mechanism to calculate the aggregated influence from the previous events on the current one. We implemented SPRITE and HP with TensorFlow (Abadi et al., 2016). We used the original implementation of NeuralHP (`https://github.com/HMEIatJHU/neurawkes`) and SAHP (`https://github.com/QiangAIResearcher/sahp_repo`), and a high-quality open-source implementation of RMTPP (`https://github.com/woshiyyya/ERPP-RMTPP`). For SPRITE and HP, we set the mini-batch size of the event sequences to 16 and learning rate $10^{-3}$. We set the dimension of the embeddings to $4$ for SPRITE. The nonlinear feature mappings $\phi_g$ and $\phi_\eta$ in (8) were both chosen as a single-layer feed-forward NN, with 16 neurons and leaky RELU as the activation function. We used the default settings of all the other methods. We ran each method with 50 epochs (enough for convergence). To avoid an unfair comparison caused by overfitting (especially for RNN based methods), we evaluated the likelihood of a validation dataset after each epoch and stopped training if there is no improvement (early stopping).

Fig. 1 a,c,e (left) show the rate function estimations of all the methods for single-type events. As we can see, the estimations of SPRITE nearly always overlap with the ground-truth rate function, demonstrating that our method can perfectly adapt to all the cases and capture different time decays. By contrast, SAHP and RMTPP's estimations significantly deviate from the ground-truth. SAHP often severely over-estimates and under-estimates the rate (see Fig. 1c and e). RMTPP correctly captures the trends, but often mistakenly estimates the local details. Although this test favors HP since only the excitation effects were considered, HP still resulted in apparent deviations in the case of quadratic and linear decays (see Fig. 1a and c), showing the limitation of the rigid exponential-decay assumption. Note that the estimation of HP overlaps with SPRITE and the ground-truth for exponential decays (see Fig. 1e). While NeuralHP also gives excellent rate estimations, in a few

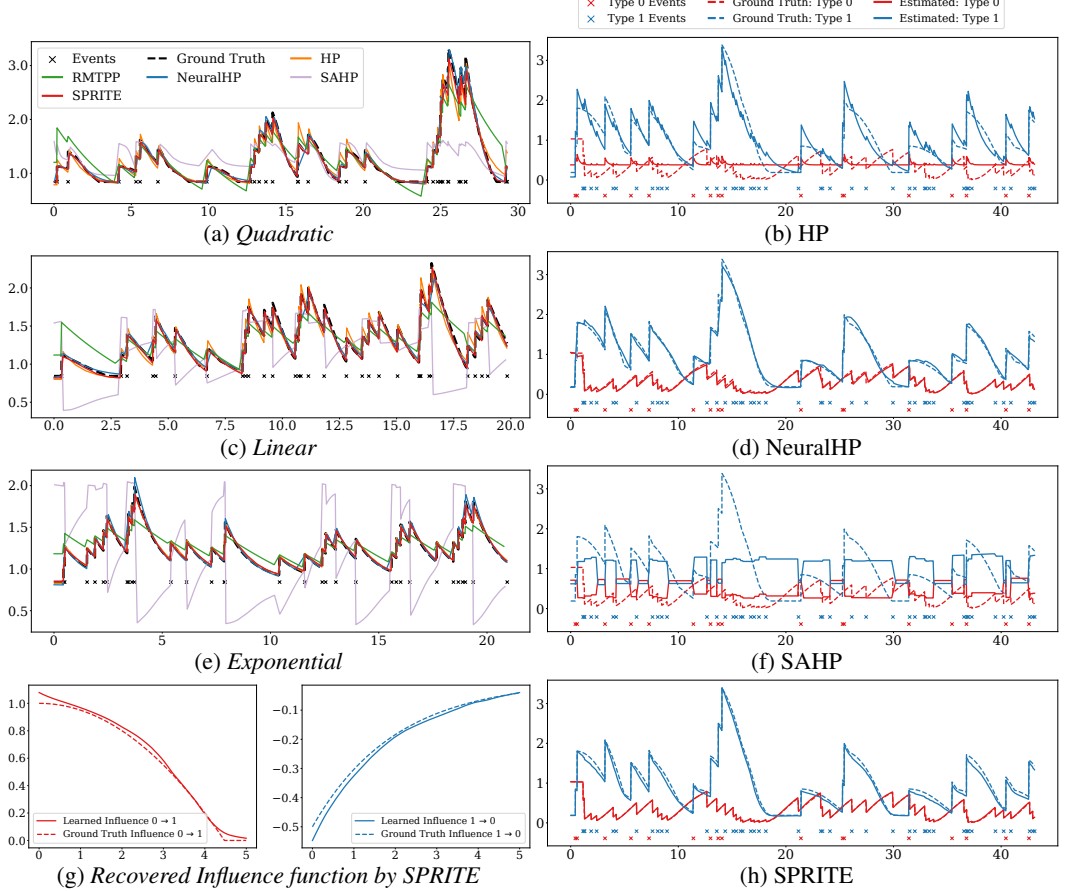

Figure 1: Rate function estimation for single-type events (a,c,e) and for bi-type events (b, d, f, h), and event influence function estimated by SPRITE for bi-type events (g). The estimated influences for single-type events are given in Fig. 1 of the Appendix. In all figures, the x-axis is time and y-axis the rate or influence value.

local regions, it still fails to capture the detail and/or gives a worse estimation than SPRITE, *e.g.*, $t \in [16, 17]$ in Fig. 1a, $t \in [4.0, 4.5]$ and $[10.5, 11.5]$ in Fig. 1c, and over-estimations for $t$ around 4.0 and 19.0 in Fig. 1e.

Fig. 1 b,d,f,h (right) show the rate estimations for bi-type events. We did not compare with RMTPP, since it only produces an overall rate function. As shown in Fig. 1b, HP's estimation for event type 0 is completely wrong, because HP cannot capture the inhibition effects from type 1 events. In Fig. 1f, while SAHP indeed captures both the triggering and inhibition effects, its rate estimations severely deviate from the ground-truth, and miss capturing many local details (see the flat curves in many intervals). Finally, NeuralHP and SPRITE can accurately estimate the rates for both events. The estimation of SPRITE for event type 0 is slightly better, because it completely overlaps with the ground-truth. Note that, however, SPRITE can further tell the influence type and strength between every pair of the events while NeuralHP, based on a blackbox LSTM, cannot. All these results have demonstrated the advantage of our method in flexibly capturing various time decays from data.

We also examined the estimated influence function by SPRITE. Fig. 1g shows the results for bi-type events. We can see that our learned decay is very accurate, nearly overlapping with the ground-truth. While the two influence functions ($h_{0 \to 1}$ and $h_{1 \to 0}$) are very different — one is convex and the other concave, they both can be accurately captured by SPRITE. The estimations for the single-type events are given in Fig. 1 of the Appendix, showing SPRITE accurately recovered all the three types of influences. The results demonstrate the power of our nonparametric model. Note that due to the blackbox nature, we cannot discover event influence functions from NeuralHP, SAHP and RMTPP.

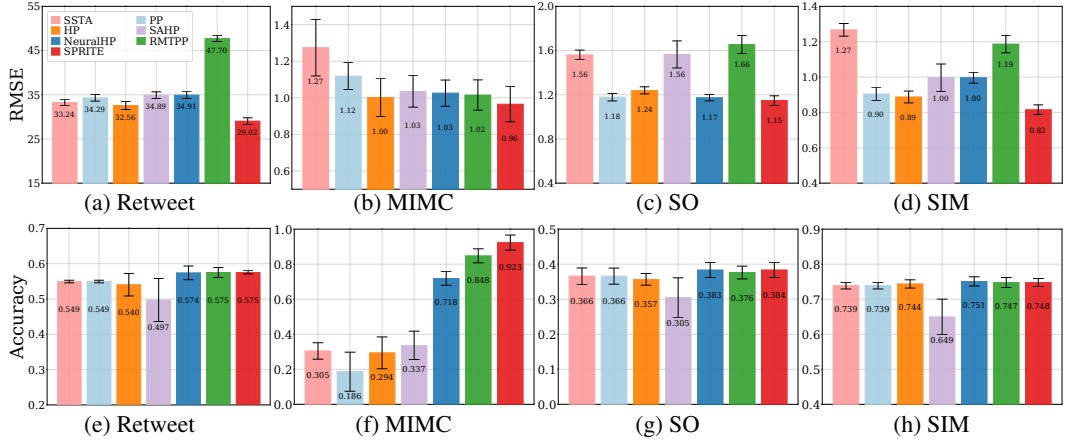

Figure 2: Prediction accuracy of the time (top row) and type (bottom row) of the future events.

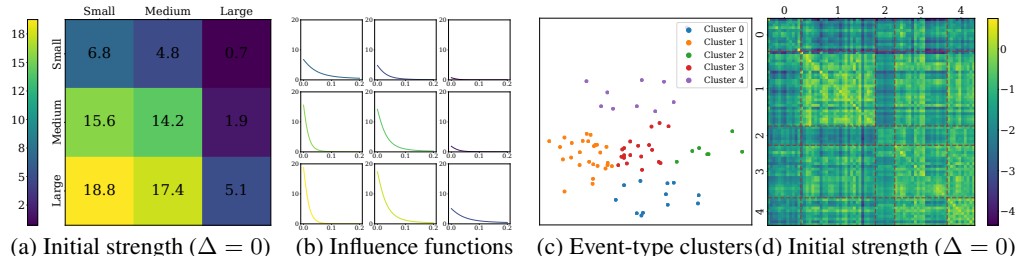

(a) Initial strength ($\Delta = 0$)    (b) Influence functions    (c) Event-type clusters    (d) Initial strength ($\Delta = 0$)

Figure 3: The learned time-decaying influences between every two types of events ("small","medium", "large") from Retweet dataset (a,b). The clustering structure of the event type embeddings and the initial strength of the influence functions organized in the event type clusters on MIMIC dataset (c, d).

## 6.2 Predictive Performance

Next, we examined the performance of SPRITE in predicting both the occurrence and type of the future events. To this end, we used the following real-world benchmark datasets. (1) Retweets (Zhao et al., 2015), $24,000$ retweet event sequences from `twitter.com`. The events were generated by three types of users, "small" retweeters with less than $120$ followers, "medium" retweeters having the number of followers between $120$ and $1,363$, and "large" retweeters with more than $1,363$ followers. The time of the first event in each sequence is labeled as $0$, and the time of the subsequent events are the time spans from the first event. (2) MIMIC (Du et al., 2016), clinical visit sequences of $650$ anonymous patients in a seven-year period. Each visit is considered as an event, and the event type is the diagnosis result. We have in total $75$ event types. (3) SO (Du et al., 2016), $6,633$ awarding event sequences in the question and answer site Stack Overflow. The users can be awarded based on both their questions and answers. There are $22$ types of events, corresponding to $22$ awards, including "Nice Question", "Good Answer", "Guru", "Great answer", *etc*. We downloaded the preprocessed datasets from `https://drive.google.com/drive/folders/0BwqmV0EcoUc8UklIR1BKV25YR1U`. In addition, we tested the prediction accuracy on (4) SIM — synthetic data of bi-type events in Sec. 6.1.

In addition to the methods mentioned in Sec. 6.1, we compared with two extra baselines: (1) Simple Statistics (SSTA) that use the average span between consecutive events to predict the future event time, and the most frequent event type to predict the future event type, and (2) a homogeneous Poisson process (PP) that assumes the event rate is a time-invariant constant.

For Retweets, MIMIC and SO, we randomly split the dataset into $70\%$ for training, $10\%$ for validation, $20\%$ for testing. We used each method to predict when the last event in a test sequence occurred and what the type is. We calculated the root-mean-square-error (RMSE) of the time predictions, and classification accuracy of the type predictions. We chose the dimension of embeddings as the 4th root of the number of event types (the rule of thumb suggested by Google (TensorFlowTeam, 2017)).

We repeated the experiments for 5 times with the 5-fold cross validation splitting scheme, and report the mean and standard deviation of RMSE and accuracy. For SIM, we randomly generated 10K sequences for training, 1K for validation, and 2K for testing in each experiment. The results of all the methods are shown in Fig. 2. As we can see, SPRITE outperforms all the competing approaches in all the cases, except that in SIM, the prediction accuracy of event types is slightly worse than NeuralHP. In many cases, SPRITE outperforms SHAP, RMTPP and/or NeuralHP by a large margin, *e.g.*, Fig. 2a, c, d, and f. The results show the advantage of SPRITE in predictive performance.

## 6.3 Pattern Discovery

Finally, we looked into the patterns discovered by our method. We first examined the learned influence functions from Retweet dataset (see (5)). In Fig. 3, we show the initial strength ($\Delta = 0$) and the curve of the influence function for every pair of event types. We found interesting patterns. First, among retweeting events are only excitation effects, and there are no inhibition effects. Second, "small" users retweeting can hardly cause or excite "large" users to retweet (the same message); the initial strength of "small $\rightarrow$ large" is 0.7 (Fig. 3a). By contrast, "large" users' retweeting can strongly incur "small" users to retweet (the initial strength of "large $\rightarrow$ small" is 18.8). This is reasonable, because "large" users have a great many of followers, and most of these followers are "small" users. The result is also consistent with the fact that "large" users, especially twitter celebrities, often act as hub nodes, and play a key role in information or opinion dissemination. It is interesting to see, though, "large" users have much less influence on other "large" users. Third, we can see across different pairs of event types, the influence varies quite much (Fig. 3b), implying the need of flexible models for their estimation. While approximating them as some exponential decays in standard HPs seems feasible, it leads to much worse prediction accuracy as compared with SPRITE (see Fig. 2a and e).

Next, we examined if our method can discover hidden structures. To this end, we investigated the learned event type embeddings from MIMIC dataset. We first performed Principled Component Analysis (PCA) to project the embeddings onto a plane, and then ran the k-means algorithm to find potential grouping structures. We used the elbow method (Ketchen and Shook, 1996) to determine the cluster number. As we can see from Fig. 3c, the (projected) embeddings reflect a clear structure within the event types. This is reasonable, because the event types are represented by the diagnosis results of the medical visits and there can be strong associations within those diagnosis results. Note that since the dataset is completely anonymized, we cannot verify the meaning of these clusters. Furthermore, we investigate the (initial) influence strengths between the event types within the same cluster and across different clusters. From Fig. 3d, we can see clear patterns. For examples, the types of events in Cluster 4 mostly excite each other, but inhibit those of Cluster 0; the types of events in Cluster 0 nearly inhibit those across all the clusters, including themselves. We also found interesting structures of the embeddings and influence functions estimated from SO dataset. We leave the results in the Appendix.

## 7 Conclusion

We have developed SPRITE, a self-adaptable point process that not only can explicitly estimate the influence type and strength between every pair of events, but also is flexible enough to capture a variety of influence time decays from data.

## Acknowledgments

This work has been supported by NSF CAREER Award IIS-2046295 and NSF IIS-1619287.

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
