# Appendix

## 1 Proof of Lemma 3.1

**Lemma 3.1.** $h \cdot \frac{\partial h}{\partial \Delta} < 0$ if and only if $h$ has the form

$$h(\mathbf{u}_{s_i}, \mathbf{u}_s, \Delta) = g(\mathbf{u}_{s_i}, \mathbf{u}_s)(1 - f(\mathbf{u}_{s_i}, \mathbf{u}_s, \Delta)), \tag{1}$$

where $g(\mathbf{u}_{s_i}, \mathbf{u}_s) = h(\mathbf{u}_{s_i}, \mathbf{u}_s, 0)$, $0 \le f(\cdot) < 1$, $\frac{\partial f}{\partial \Delta} > 0$, and $f(\mathbf{u}_{s_i}, \mathbf{u}_s, 0) = 0$.

*Proof.* First, for notational convenience, we omit the embeddings and write $h(\Delta) \overset{\text{def}}{=} h(\mathbf{u}_{s_i}, \mathbf{u}_s, \Delta)$ and $f(\Delta) \overset{\text{def}}{=} f(\mathbf{u}_{s_i}, \mathbf{u}_s, \Delta)$.

If $h$ has the form (1), we have $\frac{\partial h}{\partial \Delta} = -g \cdot \frac{\partial f}{\partial \Delta}$. Hence, we obtain

$$h \cdot \frac{\partial h}{\partial \Delta} = -g^2 \frac{\partial f}{\partial \Delta}(1 - f).$$

Since $0 \le f < 1$ and $\frac{\partial f}{\partial \Delta} > 0$, we have $h \cdot \frac{\partial h}{\partial \Delta} < 0$.

Now, from the other direction, suppose we know $h \cdot \frac{\partial h}{\partial \Delta} < 0$. To prove that $h$ must have the form (1), we first show that

$$h(0)h(\Delta) > 0 \quad (\forall \Delta \ge 0), \tag{2}$$

namely, $h(\Delta)$ always keeps the same sign with $h(0)$. Note that $g = h(0)$. We use the proof by contradiction. Specifically, suppose $h(0) > 0$ and $\exists \Delta > 0$ such that $h(\Delta) < 0$. According to the continuity of $h$, there must exist $t \in (0, \Delta)$ such that $h(t) = 0$ and $\forall s \in (t, \Delta]$, $h(s) < 0$. Then we can represent $h(\Delta)$ with its first-order Taylor expansion,

$$h(\Delta) = h(t) + h'(\xi)(\Delta - t) = h'(\xi)(\Delta - t) < 0,$$

where $\xi \in (t, \Delta)$ (according to the mean value theorem, $\xi$ is in the open interval). Since $\Delta - t > 0$, we must have $h'(\xi) < 0$. However, since $h(\xi) < 0$, we have $h(\xi)h'(\xi) > 0$, which is a contradiction. We can derive a similar contradiction when $h(0) < 0$. This proves (2). Combining the fact that $h(t)h'(t) < 0$, we immediately show that $\forall t \ge 0$,

$$h(0)h'(t) < 0. \tag{3}$$

Next, we use the fact that

$$h(\Delta) = h(0) + \int_0^\Delta h'(t)\mathrm{d}t = h(0)\left(1 + \int_0^\Delta \frac{h'(t)}{h(0)}\mathrm{d}t\right)$$

$$= h(0)\left(1 - \int_0^\Delta -\frac{h'(t)}{h(0)}\mathrm{d}t\right). \tag{4}$$

We can set $f(\Delta) = -\int_0^\Delta \frac{h'(t)}{h(0)}\mathrm{d}t$. Note that $g = h(0)$. According to (3), we have $f(\Delta) \ge 0$ and $\frac{\partial f}{\partial \Delta} > 0$. According to (2) and (4), we must have $1 - \int_0^\Delta -\frac{h'(t)}{h(0)}\mathrm{d}t = 1 - f(\Delta) > 0$. Hence $f(\Delta) < 1$. Obviously, $f(0) = 0$. $\square$

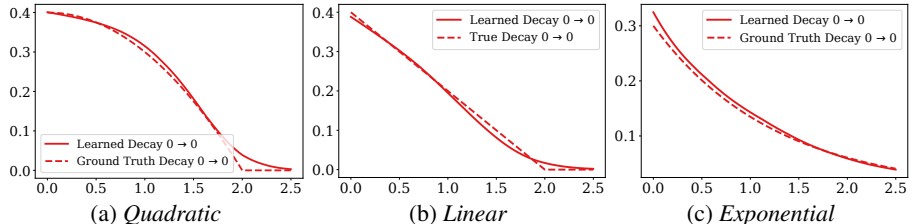

Figure 1: Estimated event influence function by SPRITE for simulation data of single-type events.

## 2 Details about Gauss Legendre Quadrature

The standard Gauss Legendre quadrature assumes the integration interval is $[-1, 1]$,

$$\int_{-1}^{1} f(x)\mathrm{d}x \approx \sum_{j} \omega_j f(\xi_j),$$

where $\{\omega_j\}$ and $\{\xi_j\}$ are quadrature weights and nodes. For our model, we want to compute the integral between $[0, \Delta]$. Hence, we first perform an integral transformation and then apply the Gauss Legendre quadrature,

$$\int_{0}^{\Delta} f(\mathbf{x})\mathrm{d}x = \int_{-1}^{1} \frac{\Delta}{2} f\left(\frac{\Delta}{2}x + \frac{\Delta}{2}\right) \mathrm{d}x \approx \sum_{j} \omega_j \frac{\Delta}{2} f(\frac{\Delta}{2}\xi_j + \frac{\Delta}{2}). \tag{5}$$

## 3 Pattern Discovery on SO dataset

We also examined the learned embeddings from SO dataset, which correspond to 22 event types. As we did on MIMIC dataset, we first performed PCA and then ran the k-means algorithm to identify the cluster structure. As we can see from Fig. 2a, the (projected) embeddings exhibit a clear structure. We then show the initial influence strengths between the event types within the same cluster and across different clusters in Fig. 2b. As we can see, there are very interesting patterns. For example, the type of events in Cluster 2 inhibit each other quite strongly, while the event types in the other clusters mostly excite each other. A few types of events in Cluster 1 are always inhibited by all types of events. Finally,we report a subset of learned influence functions in Fig. 3, including both excitation and inhibition. We can see that they obviously exhibit different time-decaying patterns, which demonstrate the flexibility of our model in estimating nonparametric time decays.

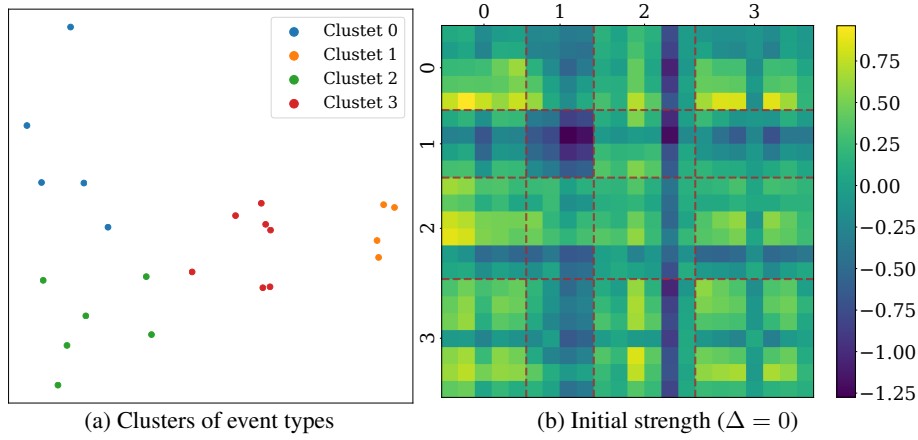

(a) Clusters of event types

(b) Initial strength ($\Delta = 0$)

Figure 2: The clustering structure of the event type embeddings and the initial strength of the influence functions organized in event type clusters on SO dataset.

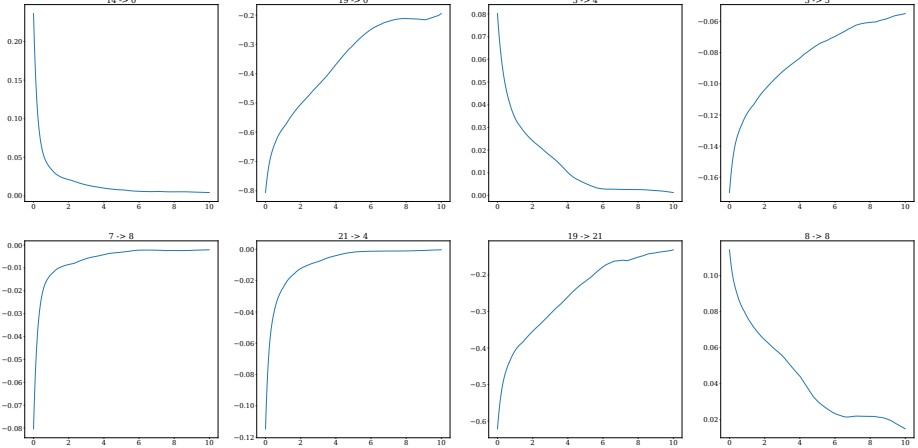

Figure 3: A subset of influence functions learned from SO dataset. The numbers on top of each sub-figure represent the event types. The x-axes are time $\Delta$, and "a->b" means the influence of events of type "a" on events of type "b".