# OpenReview forum: "Self-Adaptable Point Processes with Nonparametric Time Decays"
_NeurIPS.cc/2021/Conference — NeurIPS 2021 Poster_

### Official Review · Reviewer_k7zZ · 2021-07-14

**Rating:** 6
**Confidence:** 5

**Summary:**

The paper proposes a new algorithm to learn point process models. The proposed algorithm employs non-parametric time decay functions, which are more flexible than exciting approaches. Efficient estimation procedures are drafted. Empirical results backed up the authors claims.

**Limitations And Societal Impact:**

The authors addressed the limitations and potential negative societal impact of their work.

**Main Review:**

Originality:
The paper proposes a new solution to an important question.


Quality:
My main concern is that the proposed method is very complicated. The authors claim that existing methods make over-simplified assumptions, and propose a very complicated formulation of the time decay function (Eq. 7). The authors further place a GP prior on the formulation and propose to optimize the derived ELBO. I am not sure whether it’s worth the effort than the relatively simple approaches (e.g., RMTPP), given that the performance gain is not quite significant in some cases (Figure 2, prediction accuracy of Retweet and SO).

Clarity:
Presentation of the paper needs to improve.

1. I suggest to move line 146-147 directly after Eq. 4. It took me a while to realize that $\tilde{\lambda}$ is not the intensity function, which is confusing.

2. I suggest to move line 204-211 right after Eq. 7. Section 4.2 has nothing to do with integral estimation.

3. Some notations are not defined, e.g., $\Delta$ in Eq. 6.

4. In Lemma 3.1, any particular reasons that you choose $f(\cdot)=\mathrm{tanh}(\cdot)$ and $z(\cdot)$ to be the integral function? Are there other possible choices? The current choices seem arbitrary to me, so please provide more intuition.

Significance:
The paper studies an important problem and provides insights that could benefit future works.

**Time Spent Reviewing:**

4

---

> ### Author Response · Authors · 2021-08-09
> **Thanks for your great comments and suggestions**
>
> Thanks for your great comments and suggestions! C: comments; R: response
>
> C1: Suggestions of moving lines and defining notations.
>
> R1: Great ideas! We will follow your suggestions to improve the clarity. The $\\Delta$ in (6) is the time difference (see (4) and (5)). We will highlight its meaning in our paper.
>
>
>
> C2: “In Lemma 3.1, any particular reasons that you choose f= tanh, and z to be the integral function? Any other choices?”
>
> R2: Insightful question! We use $f = tanh$, because tanh is a natural and convenient choice that satisfies the requirement of $f$ in our lemma (i.e., $f(0) = 0, f(\\infty = 1), f’>0$). It is also very widely used. However, $tanh$ is not the only choice. We can use other functions that respect these conditions, e.g., 2* sigmoid(x) – 1. Since $tanh$ is a single-input function, we need to composite it with a function of both the embeddings and time. That’s why we introduce $z$. Intuitively, an integral function can be easily constructed to satisfy the requirement of $z$: $z(u_1, u_2, 0) = 0$, $z(u_1, u_2, \\infty) = \\infty$, and $d z/d \\Delta >0$. There might be other choices as well.  Note that, while $f$ and $z$ can have other alternatives, since $tanh$ is invertible, we can always represent $h$ as the form of (7),  i.e., Corollary 3.1.1 ($f = tanh (tanh^{-1}(f))$, where $tanh^{-1}(f)$ can be represented by the integral function chosen by us).
>
>
> C3: Concerns about the complication of the method and performance gains.
>
> R3: Great comments! We do agree that RMTPP is simple, elegant and powerful. However, like other RNN methods, the black box nature makes it difficult to extract important knowledge, such as the influence type and strength among the events. One critical goal of our work is the interpretability; our model aims to both flexibly and transparently estimate a variety of decaying influences between every pair of events. That leads to some relatively complex modeling strategies. While the prediction accuracy gains are not huge in some cases (note the gains of RMSE in Retweet, SO and SIM, of accuracy in MIMC are very significant), our method greatly improves the interpretability by recovering both the ground-truth rate and influence functions (see Fig. 1 in both the main paper and Appendix). In addition, our method can find some interesting pair-wise event influence patterns (Fig 3 of the main paper, and Fig 2, 3 in Appendix), which are hardly provided by the competing methods.

---

### Official Review · Reviewer_T9br · 2021-07-16

**Rating:** 6
**Confidence:** 4

**Summary:**

This paper proposes a multi-variate self-adaptable point process with nonparametric time decays. First, to flexibly represent arbitrary types of influences between events, they introduce an embedding to represent each event type, and model the influence between any two events as an unknown function of the event type embeddings and time span. Second, they also propose a general construction that covers all possible time decaying functions by assigning Gaussian process (GP) priors over the free latent functions and using Gauss Legendra quadrature to derive the integral in the construction. This gives their model more representative power while being computationally efficient. For efficient inference, they also use weight-space augmentation of GPs to prevent computing huge covariances matrices, and neural network feature mapping to enhance the learning capacity. Lastly, a scalable stochastic variational learning algorithm is developed by taking advantage of the re-reparameterization trick.

**Ethics Review Area:**

["I don’t know"]

**Limitations And Societal Impact:**

- The novelty of the method itself is limited and introducing Gaussian prior to the hidden embedding seems incremental.
- The representative power of this method is still limited comparing to other neural network based methods, where only excitation or inhibition effects are considered. However, the influences between events can be very complicated in some real applications. For example, an event can be inhibited in some time period, whereas it can also be excited in another time period.

**Main Review:**

- The paper is overall well-written. The problem motivation is clearly described and an efficient approach is proposed accordingly.
- Results on both synthetic and real world data are promising. The interpretable results are interesting.
- The proposed learning algorithm is efficient to carry out, which is important for point process models.

**Time Spent Reviewing:**

3

---

> ### Author Response · Authors · 2021-08-09
> **Thanks for your comments and Clarification**
>
> Thanks for your comments. C: comments; R: response
>
> C1: “The novelty of the method itself is limited and introducing Gaussian prior to the hidden embedding seems incremental.”
>
> R1: Actually, the contribution of our work is NOT “introducing a Gaussian prior over the embeddings”. **The critical contribution of our work is the nonparametric modeling of time-decaying influences between the events (inhibition or excitation; see Sec 3, Lemma 3.1 & Corollary 3.1.1).**  While the time-decaying assumption is the most commonly adopted and successful assumption in point process modeling, there is not any current work that can flexibly and transparently recover all kinds of decays; they all assume parametric forms; the most common is exp (see Eq1). Our methods, like the general construction of the influence function (see (6) and (7)), without the need for imposing derivative constraints, placing GP priors over the latent functions in the construction, and using Gaussian-Legendre for the intractable integral, are all new, and never appears in the previous work. Therefore, we believe that, as commented by Reviewer k7zZ, “The paper proposes a new solution to an important question”.
>
>
>
> C2: “The representative power of this method is still limited comparing to other neural network based methods, where only excitation or inhibition effects are considered. However, the influences between events can be very complicated in some real applications. For example, an event can be inhibited in some time period, whereas it can also be excited in another time period.”
>
> R2: First, we agree that the representation power of our method might still be less than the RNN based models. However, as emphasized by our paper (line 36-39, line 80-100), the RNN based methods are untransparent and lack interpretability --- they couple all the temporal dependencies in a black-box, and are difficult to distill meaningful/important knowledge, like influence types and strengths between pairs of events. By contrast, our method well addresses the interpretability issue while still is quite expressive (see the second point). This is the major motivation of our work, and a key advantage of our model (see Fig 1 and 3).
>
> Second, our model focuses on the time-decaying influences, the most widely used and successful modeling assumption in point processes.  Our model is flexible enough to capture all kinds of decaying influences, not restricted to any parametric form (e.g., exponential, linear, and quadratic; see Fig 1). In fact, it achieves maximum flexibility among this most important/popular family of point processes.
>
> Third, our model is actually expressive enough to capture “an event can be inhibited in some time period, whereas it can also be excited in another time period.”  Note that the influence function in Eq5 is a function of time and can be both positive and negative. The raw rate for a particular type of events (see Eq4) integrates the influences from all the previous events. Hence, it covers the case that in one time period, the total summation of the influences (from the past events) is positive, meaning the event is excited, while in another time period, the summation of the influences is negative and so the event is inhibited.
>
> Fourth, we would like to cite George Box's famous quote: “all models are wrong, some are useful.” We believe that pursuing the maximum representation power (say, 100-layer neural networks) is not always the best choice. It really depends on the problem or task. Such models can bring in other issues, like interpretability, overfitting, being hard for training, which can be incompatible with the specific task. We believe the proposed model is a good fit for the problem we want to address in our paper; as commented by Reviewer VtoS, “it addresses an important problem currently not very-well addressed by current research”.

---

> > ### Comment · Reviewer_T9br · 2021-08-23
> > **Reply to the authors' response**
> >
> > Thank you for the detailed response. Some of my concerns have been addressed. I will definitely be willing to raise my score to 6 if other reviewers all agree to the acceptance.

---

### Official Review · Reviewer_VtoS · 2021-07-16

**Rating:** 6
**Confidence:** 4

**Summary:**

The paper proposes a semi-parametric class of point processes which is intensity-based while using kernel-based,non-parametric functions to model the effects between types of events in the intensity function. This reduces the number of the learnable parameters hence making the model applicable to datasets with a large number of event types.  In particular, it

(1) formulates in Lemma 3.1 the property of such kernel functions that guarantee a time-decaying effect of the influence functions when there is present either an excitatory effect or an inhibitory effect

(2) proposes a class of kernel functions in corollary 3.1.1 . that respects the property in (1)

(3) suggests a training algorithm that leverages a bayesian linear regression model for constructing the kernel functions and stochastic variational inference on batches of types and event sequences for accelerating inference.

**Limitations And Societal Impact:**

yes

**Main Review:**

In general, I think the paper is well-written while it addresses an important problem currently not very -well addressed by current research. Its novelty lies in the design of kernel functions that exhibit time-decaying effect. The experimental results are compelling. However, some points in the model/ training algorithm are not very clear while the experiments should be extended to further demonstrate the model. In particular,

(1) the model uses a soft-plus transformation to guarantee the positivity of the intensity function while maintaining the inhibitory effects imposed by the term in Eq 4. My impression is that this transformation renders the integral in the objective function of Eq 9 not computable in closed form. How do the authors address this (monte carlo etc?)

(2) it is not very clear to me why the authors do not consider a posterior for the type embdeddings u in the final objective L after Eq (10)

(3) I haven't seen partition on event-based datasets with respect to the types of events when performing variational inference? are there convergence guarantees? i think it would help a lot, if convergence, was at least demonstrated with learning curves for some parameters of the model.

(4) It would be interesting if an ablation study was provided which shows predictive performance for different dimensionality of type embeddings R, given that it exhibits cubic time complexity?

(5) I believe that the synthetic experiments should be extended so that synthetic event sequences are generated by a Hawkes process, neural point process etc and show how well the proposed model can recover existing point process models.

(6) the real world dataset experiments would become stronger if the authors also compare with:

[1]Shchur, O., Biloš, M. and Günnemann, S., 2019. Intensity-free learning of temporal point processes. arXiv preprint arXiv:1909.12127.

[2] Mehrasa, N., Jyothi, A.A., Durand, T., He, J., Sigal, L. and Mori, G., 2019. A variational auto-encoder model for stochastic point processes. In Proceedings of the IEEE/CVF Conference on Computer Vision and Pattern Recognition (pp. 3165-3174).

**Time Spent Reviewing:**

4

---

> ### Author Response · Authors · 2021-08-09
> **Thanks for your great comments and questions**
>
> Thanks for your great comments and questions. C: comments; R: response
>
> C1: “the model uses a soft-plus transformation to guarantee the positivity of the intensity function while maintaining the inhibitory effects imposed by the term in Eq 4. My impression is that this transformation renders the integral in the objective function of Eq 9 not computable in closed form. How do the authors address this (monte carlo etc?)”
>
> R1: That’s a great question. The integral is indeed intractable. Actually, we have explained how we deal with the integral in the paper, as in line 191-193 --- “we represent the integral as an expectation” with respect to a uniform distribution, so that we can easily calculate an unbiased stochastic gradient to conduct stochastic optimization (see line 196-203). This also can be viewed as a type of MC method.
>
> C2: “It is not very clear to me why the authors do not consider a posterior for the type embeddings u in the final objective L after Eq (10)”
>
> R2: Great question and suggestion. We used point estimation for the type embeddings only for convenience, since the focus of our paper is to learn a nonparametric time-decaying influence function. We can definitely introduce another variational posterior of the embeddings to jointly optimize the ELBO. We will conduct such experiments.
>
> C3: “I haven't seen partition on event-based datasets with respect to the types of events when performing variational inference? are there convergence guarantees? i think it would help a lot, if convergence, was at least demonstrated with learning curves for some parameters of the model.”
>
> R3: Actually, each event sequence does not only contain one type of events. Each sequence consists of mixed types of events, where different types of events influence each other. The datasets include many sequences. If we partition the data according to the event type, we will break these sequences and lose the time dependencies of the events. Hence, we follow the setting of the prior works (also the competing methods), NeuralHawkes, RMTPP, SHAP, etc., to partition the data into training and test event sequences (without breaking the sequence structures).
>
> Second, since the variational model evidence lower bound (ELBO) (L under line 186) is upper bounded by the log model evidence (according to the variational learning theory), and we conduct stochastic optimization to maximize the ELBO. The convergence is theoretically guaranteed. Thanks for your suggestion, we will supplement the reference of variational inference and the learning curves of our model.
>
> C4: “It would be interesting if an ablation study was provided which shows predictive performance for different dimensionality of type embeddings R, given that it exhibits cubic time complexity?”
>
> R4: Great suggestion! We will add the examination of different embedding dimension R.  Note that R is usually chosen to be very small, e.g., 5 or 10, R<< sequence length and/or sequence number.  The time complexity of our algorithm is not dominated by R.
>
> C5: “I believe that the synthetic experiments should be extended so that synthetic event sequences are generated by a Hawkes process, neural point process etc and show how well the proposed model can recover existing point process models.”
>
> R5: Great suggestion. We do agree. The current experiment aims to demonstrate our model can indeed recover different time-decaying influences (i.e., nonparametric). Adding more experiments to see if our model can recover other point process models’ rate is very interesting. Note that, our synthetic data (1)(2)(3) (line 265-267) have already been generated by Hawkes processes, because among the events are excitation effects; the bi-type synthetic data (line 268-271) is generated by an even more general point process, since both the inhibition and excitation effects are simulated. We will supplement more experiments.
>
> C6: “the real world dataset experiments would become stronger if the authors also compare with [1] [2]”
>
> R6: Thanks much for providing these great references! We will definitely cite, discuss and compare with these excellent works.

---

### Official Review · Reviewer_jsqH · 2021-07-17

**Rating:** 8
**Confidence:** 4

**Summary:**

This work proposes a new marked temporal point process model that manages to leverage neural networks in such a way to recover and identify the influence that certain events have on future ones. This is accomplished by modeling the overall intensity function of the model as a summation of individual decaying effects that prior events have on the rate of occurrence for future events.

**Limitations And Societal Impact:**

I think that there could be a bit more discussion surrounding the societal impact of this model.

**Main Review:**

There is a definite lack of interpretability in the methods proposed by the neural point process literature. Point process models are very powerful predictive tools; however, many of the desired use cases for them require interpreting why certain events are more likely to occur given a particular history (e.g., within healthcare). From my perspective, this work makes a solid step towards addressing this issue. The proposal is well founded, has a clear motivation, and adequately demonstrates the effectiveness via empirical results.

 As for clarity, the main thrust of the paper is very clear. I do think that from about lines 180 to 200 and the entire synthetic data experiment section is a little dense and could stand to be edited for readability (I do understand that this is most likely a byproduct of the submission page limit).

I am curious about a few things that weren't directly answered in the paper:
1. What is the added effect of treating the embeddings probabilistically? Do the results differ significantly if, for instance, all of the parameters are learned deterministically without any priors? I would imagine that the results of at least 6.3 would be affected, but I wonder if this is actually the case empirically. Note that I do think the proposal is significant due to the formulation of the intensity function alone.
2. What sort of structure do $\phi_g$ and $\phi_\eta$ take on? I understand that they are neural networks, but I cannot find any details about their hyper parameters for the experiments (i.e., the number of layers they have, activation function, initial values, etc.).
3. For the experiments using real data, were the sequences truncated at all or were the entire sequences modeled during training and inference?

**Time Spent Reviewing:**

3 hours

---

> ### Author Response · Authors · 2021-08-09
> **Thanks for your valuable comments**
>
> Thanks for your valuable comments! C: comments; R: response
>
> C1: "What is the added effect of treating the embeddings probabilistically? Do the results differ significantly if, for instance, all of the parameters are learned deterministically without any priors? I would imagine that the results of at least 6.3 would be affected, but I wonder if this is actually the case empirically."
>
> R1: Thanks for the insightful question! First, the prior distribution over the embeddings has the regularization effect and therefore could prevent overfitting. Our model uses the Gaussian prior, which corresponds to L2 regularization.  Second, this will make our model fully probabilistic, which is more principled and allows for other probabilistic inference approaches, like Hamiltonian Monte-Carlo sampling (HMC). While we developed variational inference for efficiency and scalability, we hope our model can also be estimated by other standard approaches when applicable, like using HMC when data is small. Third, our experience indeed shows that the performance, especially the prediction accuracy, is better than learning the parameters “deterministically with any prior”.
>
>
>
> C2: the structure of $\\phi_g$ and $\\phi_\eta$
>
> R2: Great question. $\\phi_g$ and $\\phi_\eta$ are two single-layer feed-forward neural networks, with leaky RELU as the activation function. Our experience shows that they are indeed better than popular RBF kernels. We will supplement the setting of $\\phi_g$ and $\\phi_\eta$ in the paper.
>
>
> C3: "For the experiments using real data, were the sequences truncated at all or were the entire sequences modeled during training and inference"
>
> R3: Great question! We preserved most of the sequences but truncated a few sequences that are much longer than the remaining to prevent them from dominating/disturbing the training or testing. We will supplement these details in our paper.
>
>
> C4: “I think that there could be a bit more discussion surrounding the societal impact of this model.”
>
> R4: Thanks for this great idea and suggestion! We will definitely add such discussions to the paper. One example is the medical application, where our model can be used to discover the mutual influences within clinical visit events. This might help physicians and medical researchers to find new insights about the relationships between diagnosis, treatment, prognosis, etc. These discoveries might be useful to improve the medical service and patient experience.

---

### Decision · Program_Chairs · 2021-09-27

**Decision:**

Accept (Poster)

**Comment:**

The paper proposes a new marked temporal point process model for continuous-time event sequences. While the standard neural stochastic point process models (e.g., Neural Hawkes Processes) parameterized the time-varying intensity functions using recurrent neural networks,  these models are often black boxes and don't allow to estimate the influence (inhibition or excitation) that event types have on future ones. In contrast, the proposed work uses interpretable Gaussian processes and achieves better performance while maintaining interpretability and scalability. In this sense, the approach achieves to both flexibly and transparently estimate various decaying influences between every pair of events.

With few exceptions on the model/ training algorithm that could be presented more clearly, the paper is overall well-written and mathematically sound. Despite its sophisticated inference scheme, it provides a valuable contribution to machine learning and statistics.

Please run a grammar check over the paper for the final version since many small mistakes are present.